# Effects of Environmental Factors on the Spatial Distribution Pattern and Diversity of Insect Communities along Altitude Gradients in Guandi Mountain, China

**DOI:** 10.3390/insects14030224

**Published:** 2023-02-24

**Authors:** Lijuan Zhao, Ruihe Gao, Jiaqi Liu, Lei Liu, Rongjiao Li, Lina Men, Zhiwei Zhang

**Affiliations:** 1Department of Forest Conservation, College of Forestry, Shanxi Agricultural University, Jinzhong 030801, China; 2Shanxi Dangerous Forest Pest Inspection and Identification Center, Jinzhong 030801, China

**Keywords:** altitudinal gradient, distribution pattern, insect community, Guandi mountain

## Abstract

**Simple Summary:**

Elevation gradient is an important factor affecting insect species composition, community structure, and the spatial pattern of diversity. At present, it is necessary to study the impact of environmental factors on the insect population structure and diversity patterns along altitudinal gradients in Guandi Mountain, China. This study revealed that altitude gradient could significantly affect the composition and distribution patterns of insect communities and the insect community showed certain differentiation characteristics with the altitude gradient in this area. Additionally, the results of redundancy analysis (RDA) and correlation analysis indicated that soil physicochemical properties were closely related to the distribution and diversity of insect taxa orders along the altitude gradient, and the soil temperature was the most significant environmental factor affecting the insect community structure and diversity on the altitude gradient. Our results suggest that the interactive effects of altitude and environmental variables play an important role in determining the community structure, distribution patterns, and diversity of insect populations.

**Abstract:**

Understanding the distribution patterns and underlying maintenance mechanisms of insect species is a core issue in the field of insect ecology. However, research gaps remain regarding the environmental factors that determine the distribution of insect species along altitudinal gradients in Guandi Mountain, China. Here, we explored these determinants based on the distribution pattern and diversity of insect species from 1600 m to 2800 m in the Guandi Mountain, which covers all typical vegetation ecosystems in this area. Our results showed that the insect community showed certain differentiation characteristics with the altitude gradient. The results of RDA and correlation analysis also support the above speculation and indicate that soil physicochemical properties are closely related to the distribution and diversity of insect taxa orders along the altitude gradient. In addition, the soil temperature showed an obvious decreasing trend with increasing altitude, and temperature was also the most significant environmental factor affecting the insect community structure and diversity on the altitude gradient. These findings provide a reference for exploring the maintenance mechanisms affecting the structure, distribution pattern, and diversity of insect communities in mountain ecosystems, and the effects of global warming on insect communities.

## 1. Introduction

Insects, as an important component of terrestrial ecosystems, have a strong ability to adapt to the environment and play an important role in nutrient circulation and energy transformation [1]. Insects are a diverse group with differing habits and play an important role in maintaining the ecological balance of different ecosystems. Additionally, they have a significant impact on forest ecosystem processes and service functions [2,3]. Furthermore, insects are important indicator organisms that are highly sensitive to changes in climate, vegetation, soil, and other environmental factors [4].

Exploring the distribution patterns and maintenance mechanisms of insect species is a core issue in the field of insect ecology [5]. Numerous studies have shown that the distribution and diversity of insect species are determined by complex biotic and abiotic factors, which mainly include climatic factors, plant communities, soil characteristics, and elevation gradients [6,7,8,9,10,11,12]. A large number of studies have shown that abiotic factors, such as temperature and relative humidity, are key factors determining insect diversity and species distribution and can significantly change the species composition of insect community structures [13]. In the context of global ecological change, a large number of studies have shown that global climate change may significantly alter insect community structures, and warmer temperatures will affect the activity and survival of insects [14,15,16]. When the soil nutrient content is higher in a forest ecosystem, it may promote the growth of plants to a certain extent [12,17], which in turn increases the species and number of insect populations [18,19,20]. However, due to the differences in environmental variables, such as climate, topography, soil, and vegetation types, it is difficult to reach a consensus on the impact of environmental variables on insect communities in different regions. Hence, the maintenance mechanisms that influence insect community structure, distribution patterns, and diversity in different ecosystems should be explored further.

Elevation gradients can condense a large number of variations in environmental factors into a small geographical space, which provides useful natural experimental conditions for studying biodiversity distribution [21,22,23,24]. As ectotherms, insects are very sensitive to changes in environmental temperature; therefore, slight changes in the temperature can directly affect their development, reproduction, and survival [6]. Therefore, the elevation gradient is an important factor affecting insect species composition, community construction, and the spatial pattern of diversity [23,25]. Studies on altitudinal patterns of insect community species diversity in forest ecosystems are helpful in revealing the status of global biodiversity and its maintenance and change mechanisms [26]. Additionally, many studies have shown that elevation has the same effect on vegetation and soil properties as latitude does [27,28,29,30].

The Guandi Mountain is located in the middle range of the Lvliang Mountains in the Shanxi Province, with an elevation gradient variation ranging from 1600 to 2831 m. The vegetation types in Guandi Mountain show obvious vertical zonation along the altitude gradient, and the forest vegetation types in this area are relatively complete in northern China. The forest vegetation from low altitude to high altitude is as follows: *Quercus wutaishansea* (Mary.) forest (1600–1700 m), *Pinus tabulaeformis* (Carr.) forest (1600–1800 m), subalpine poplar–birch (*Populus davidiana* Dode. and *Betula platyphylla* Suk.) mixed forest (1700–2100 m), secondary spruce forest characterized by *Picea wilsonii* (Mast.) and *P. meyeri* (Rehd. et Wils.) (1900–2500 m), bright coniferous forest dominated by *Larix principis-rupprechtii* (Mayr.) (1800–2600 m), and sub-alpine meadow (>2600 m) [31]. At present, studies on the changes in insect community structure, distribution pattern, and diversity with altitude gradient in Guandi Mountain are limited. Hence, it is necessary to study the impact of environmental factors on the insect population structure and diversity patterns in this area.

In this study, we selected seven typical vegetation community ecosystems at different elevation gradients: including *Quercus wutaishansea* forest (QWF), *Pinus tabulaeformis* forest (PTF); *Populus davidiana* and *Betula platyphylla* mixed forest (PBM); *Picea wilsonii* forest (PWF), *P. wilsonii*, and *Larix principis-rupprechtii* mixed forest (PLF); *L. principis-rupprechtii* forest (LPF); and sub-alpine meadow (SAM), covering all the typical ecosystems in the Guandi Mountain. The main objectives of this study were to investigate the distribution patterns and diversity of insect species from 1600 m to 2800 m in the Guandi Mountain. Specifically, this study aimed to address the following three scientific questions: (1) whether insect community structure and composition changes significantly at different altitude gradients; (2) what is the variation pattern of alpha diversity of insect species along the altitude gradient; and (3) which environmental variables influence the spatial distribution and diversity of insect communities along altitudinal gradients.

## 2. Materials and Methods

### 2.1. Study Sites

This study was conducted in the Guandi Mountain, Shanxi Province, China (37°20′–38°20′ N, 110°18′–111°18′ E). This area is characterized by a warm temperature continental climate, with an annual average temperature of 3.5 °C and annual precipitation of 830 mm [31]. In addition, the soil types present a vertical distribution belt, and from bottom to top, they are light brown soil, mountain brown soil, mountain eluviated brown soil, and sub-alpine meadow soil.

Seven different ecosystems were selected to investigate the composition and spatial distribution patterns of insect communities along the altitude gradient in the Guandi Mountain. Three 20 × 20 m standard plots were set for each typical ecosystem as repetitions, and 21 standard plots were investigated in this study (Table 1 and Appendix A, values are mean ± SD). From July to August 2021, the “Tally” method was used to survey detailed information of woody plants with diameter at breast height (DBH) larger than 2.5 cm in each plot, including the name of tree species, DBH, tree height, and the size of crown [32]. Geographic coordinates, altitude, slope, canopy density, and other basic information were recorded for each plot.

### 2.2. Insect Sampling and Specimen Identification

In this study, the methods of “sweep net sampling” and “pitfall trapping” were used to collect insect specimens in different altitude vegetation ecosystems [33]. Insect specimens were collected every seven days, and the survey time was concentrated from early July to the end of August in 2020 and 2021. The “sweep net sampling” method was used to collect low flying insect species, and the net was swept more than 200 times in each plot [32]. According to the “pitfall trapping” method (Figure 1), five transparent plastic containers with a diameter of 10 cm and a height of 12 cm were placed at the center and the four end points in each plot. Each small container was buried in the soil, and the container was level with the ground surface. Approximately 75 mL of sweet and sour alcohol mixture was poured into each container (Appendix A), in which the ratio of the mixture was brown sugar:vinegar:water:75% alcohol = 3:4:2:1. All collected insect specimens were stored in a 75% alcohol solution for preservation. Additionally, all collected insect were classified morphologically as accurately as possible into genera and species according to relevant professional books and reference materials [34,35,36,37,38,39,40]. The COI barcoding can be used for molecular identification of insect species that are difficult to identify morphologically [41].

### 2.3. Environmental Data Collection 

The “soil core sampling” was used to determine the physical and chemical properties of the 0–10 cm soil layer samples in each plot. First, the thickness of the litter layer was recorded, and subsequently the temperature, relative humidity, conductivity, and pH of the soil layer were measured using a handheld soil parameter meter. Soil layer samples were collected by the “cutting ring method” (ring = 100 cm^3^) and then taken to the laboratory to be measured for water-related physical properties, including soil bulk density (BD, g/cm^3^), maximum water holding capacity (MWHC, %), capillary water holding capacity (CWHC, %), capillary porosity (CP, %), noncapillary porosity (NP, %), and total soil porosity (TSP, %) [7,42]. To determine the chemical properties, samples of the 0–10 cm soil layer were taken back to the laboratory and impurities such as rocks, roots, and animal and plant residues were removed. Following this, the samples were dried naturally, ground, and passed through a 2 mm soil sieve. The Kjeldahl nitrogen method was used to determine the total nitrogen content of each sample, and a UV-Vis spectrophotometer (UV-2550, Shimadzu, Kyoto, Japan) was used to analyze the content of available phosphorus (%) and available potassium for each soil sample.

### 2.4. Data Analysis

All environmental variables and insect metrics were compared using one-way analysis of variance (ANOVA) and Fisher’s least significant difference (LSD) tests with an alpha value of *p* < 0.05, using SPSS 22.0 (SPSS Inc., Chicago, IL, USA). Principal component analysis (PCA) was used to examine the differentiation characteristics of insect communities along the altitude gradient using a free online platform for data analysis (https://www.genescloud.cn (accessed on 15 February 2023)). To compare the variation in insect diversity along the altitudinal gradient, the insect Hill numbers of each plot were calculated [43,44,45,46], including species richness (S), Shannon–Wiener diversity (H’), Inverse Simpson’s index (1/D), and Berger–Parker index (1/d). Heatmaps (*n* = 21) for insect taxa order along the altitudinal gradient were plotted using heatmap tools on the Genescloud platform (https://www.genescloud.cn (accessed on 15 February 2023)). Statistical analyses of environmental variables along the altitudinal gradient were performed using SPSS 22.0 and plotted using GraphPad Prism 6.0 (GraphPad Software, La Jolla, CA, USA). Pearson correlation coefficients were used to test the correlation of environmental variables with insect taxa order and diversity using a free online platform for data analysis (https://www.genescloud.cn (accessed on 15 February 2023)). The significance level of the correlation was set at *p* < 0.05. In order to analyze the ordination relationship between environmental variables and insect communities at different altitude gradients, the redundancy analysis and mapping of environmental variables that affected insect community structure and diversity were performed using CANOCO 5.0 (Microcomputer Power, Ithaca, NY, USA).

## 3. Results

### 3.1. Variations of Environmental Variables along the Altitudinal Gradient 

The soil physicochemical properties varied greatly among different elevations (Appendix A), and the variation trend of each soil factor differed along the altitudinal gradient (Figure 2). The soil pH in the subalpine meadow at 2800 m was acidic, and the soil temperature, humidity, and EC values were significantly higher than those in the forest ecosystems. In the forest ecosystem in the 1600–2600 m elevation, the values of soil temperature, relative humidity, and EC all showed a significant decreasing trend, but soil pH showed a rising trend. In addition, the soil nutrients of available N, P, and K showed a slight decreasing trend with an increase in the elevation gradient. However, the N/P, N/K, and P/K ratios were relatively stable and the differences among groups with different elevation gradients did not reach a significant level. Overall, the soil bulk density showed an obvious upward trend with an increase in the elevation gradient, and the differences between the groups were statistically significant (*p* < 0.05). In contrast, other soil water-related physical properties, including the maximum water holding capacity, capillary water holding capacity, capillary porosity, and total soil porosity, showed an obvious downward trend with increasing altitude gradient. 

### 3.2. Variations in Insect Community Composition along the Altitudinal Gradient 

A total of 9321 individuals from 11 orders, 80 families, and 221 species were collected in this study (Table 2 and Appendix A). The dominant insect groups were Coleoptera, Diptera, Orthoptera, Hemiptera, and Hymenoptera, which accounted for 87.64%, 94.96%, and 97.61% of the total at the family, species, and individual levels, respectively. Other insect groups accounted for less than 5% of the total insect population at the family, species, and individual levels. 

PCA showed that the insect community had distinct differentiation characteristics along the altitude gradient (Figure 3). Overall, insect populations and the number of individuals in the subalpine meadow ecosystem at 2800 m were significantly higher than those in the forest ecosystems from 1600 to 2600 m. Additionally, there was a downward trend in the insect population at the levels of order, family, species, and individuals with increasing of elevation. The number of individual insect species decreased significantly, and the difference between groups reached a significant level (Figure 4).

Cluster analysis revealed that insect groups in vegetation community ecosystems were clustered into different categories along the elevation gradient (Figure 5), indicating that altitude significantly affected the composition and distribution of insect communities. From the heatmap, it is clear that the dominant taxa in the typical vegetation from 1600 to 2400 m were Diptera, Coleoptera, and Hymenoptera. However, populations of Hemiptera and Orthoptera gradually became dominant when the altitude exceeded 2600 m.

The Hill numbers of insect composition and diversity along the altitudinal gradient are shown in Table 3. One-way ANOVA and LSD tests demonstrated that Species richness (*p* = 0.000) was significantly affected by altitudinal gradient, while there was no significant difference in Shannon–Wiener index (*p* = 0.07), Inverse Simpson’s index (*p* = 0.29), and Berger–Parker index (*p* = 0.21).

### 3.3. Correlation of Environmental Variables with Insect Community and Diversity

The relationship between insect communities and soil properties was analyzed using Pearson correlation coefficients (Figure 6). The Coleoptera insect group was positively correlated with soil BD (*p* < 0.05). The Diptera insect groups were significantly positively correlated with soil temperature (*p* < 0.05) but had a significantly negative correlation with soil relative humidity (RH) and P/N (*p* < 0.05). The Hemiptera insect groups were positively correlated with soil electrical conductivity (EC) and P/N (*p* < 0.05) and significantly positively correlated with soil AN, AP, and AK (*p* < 0.01), whereas they had a significantly negative correlation with soil pH (*p* < 0.001). The Thysanoptera insect groups had a positive correlation with soil temperature (*p* < 0.05), and the Lepidoptera insect groups had a negative correlation with soil RH (*p* < 0.05). The content of AN, AP, and AK in the soil significantly affected the Neuroptera insect groups. The Mantodea insect groups showed a positive correlation with the ratio of soil AP to AN (*p* < 0.05) and a negative correlation with soil pH (*p* < 0.05). 

Moreover, soil temperature, pH, EC, AN, AP, AK, BD, and P/N were also closely related to insect diversity (Figure 7). The number of insect individuals was positively correlated with soil BD (*p* < 0.05) and significantly positively correlated with soil temperature (*p* < 0.001). Both the number of insect species (S) and Pielou evenness index (J) were positively correlated with soil temperature, EC, AN, AP, AK, BD, and P/N and were significantly negatively correlated with soil pH (*p* < 0.001). 

### 3.4. Effects of Environmental Variables on the Spatial Distribution and Diversity of Insect Communities along Altitudinal Gradients

To further understand the main soil factors affecting the distribution of insect communities along the altitude gradient, redundancy analysis was performed to correlate the soil variables and insect taxa orders (Figure 8). All soil environmental variables accounted for 56.09% of the canonical eigenvalues in the insect groups with different elevation gradients. Insects with different altitude gradients had obvious clustering characteristics and showed different directions in the ordination biplot. Moreover, soil temperature was the major soil environmental variable influencing the distribution of insect groups along the altitude gradient. As an important insect community in the forest ecosystem at an altitude of 1600 m, the distribution of the Diptera and Coleoptera communities was positively correlated with soil temperature and negatively correlated with soil RH. The distribution of Hemiptera and Orthoptera along the altitude gradient was positively correlated with soil BD, EC, AN, AP, and AK, but negatively correlated with soil pH, MWHC, CWHC, NP, CP, and TSP. Additionally, increases in soil pH, MWHC, CWHC, NP, CP, and TSP may promote an increase in the Hymenoptera population. The soil properties selected in this study had a relatively marginal effect on the population distribution of Dermaptera, Neuroptera, Thysanoptera, Mantodea, and Lepidoptera.

## 4. Discussion

Elevation gradients are optimal ecological surrogates for inferring global change-driven dynamics and can minimize the confounding effects of historical and biogeographic differences in species pools [6,8,47]. Elevation integrates the gradient effects of various environmental factors such as light, temperature, humidity, and precipitation [21,23,25,48]. Insects are widely used as indicators to evaluate ecosystem biodiversity and for environmental assessment, mainly because they are relatively small in size, widely distributed, inhabit complex and diverse environments, and are very sensitive to environmental changes [2,3,4]. Therefore, understanding the changes in insect community composition, distribution patterns, and diversity with altitude gradients is helpful to reveal the current status of global biodiversity and its maintenance and change mechanisms.

The results of this study showed that the altitude gradient could significantly affect the composition and distribution patterns of insect communities in Guandi Mountain. Among the 9321 insects obtained in this study, the dominant insect groups were Coleoptera, Diptera, Orthoptera, Hemiptera, and Hymenoptera. PCA showed that the insect community had obvious differentiation characteristics along the altitude gradient. Specifically, as the elevation increased from 1600 m to 2600 m, the abundance of insects in the forest ecosystem showed a declining trend at the order, family, species, and individual levels. This is mainly due to the comprehensive influence of hydrothermal conditions, microtopography, and soil texture changes, which tend to reduce insect species and abundance, and mainly show a monotonously decreasing distribution pattern along the altitudinal gradient [49]. However, the distribution pattern of insect species is not absolute with increasing altitudinal gradients. Previous studies have shown that species density generally shows more than a unimodal peak decreasing trend with increasing elevation [27,28]. Other studies have also shown that altitude is a key factor affecting the distribution pattern and diversity of insect populations, and that the distribution pattern of insects generally presents a monotonously increasing, monotonously decreasing, and multi-peak change pattern with increasing altitude [50,51,52]. Therefore, it is difficult to reach a consensus on the change patterns of insect communities along the altitudinal gradient among different regions, which is mainly due to differences in altitudinal range, climate factors, soil physicochemical properties, and vegetation types [21,22,23,24,26].

Consistent with the results of numerous previous studies, this study showed that the composition and diversity of insect groups had high spatial variations among the seven selected vegetation community ecosystems along altitude gradients. The results clearly showed that the dominant groups in the typical vegetation from 1600 to 2400 m were Diptera, Coleoptera, and Hymenoptera. However, the populations of Hemiptera and Orthoptera gradually became dominant when the altitude exceeded 2600 m. These phenomena may be caused by differences in vegetation community composition and soil physical and chemical properties along the elevation gradient of the Guandi Mountain [17,22,23,24,26]. To a certain extent, the insect community was significantly affected by the plant community in a particular forest ecosystem, and the richness and diversity of the insect community were higher in the ecosystem with more complex plant community structure and higher diversity [9,53]. Indeed, higher tree richness and density can increase the niche space available for herbivorous insects in forests [54,55]. Many studies have shown that the insect community populations and diversity are higher in forest ecosystems with higher vegetation diversity, and different mixing proportions of forest tree species have a significant effect on the distribution of insect populations [9,56,57]. Moreover, previous studies have shown a positive correlation between insect species richness and plant density and that the high abundance of plant species found in tropical forests is responsible for the great diversity of insect species [58,59,60]. In this study, because of the relatively high richness and coverage of vegetation species and complex forest structure types in lower altitude forest ecosystem, the composition and structure of insect communities can be expected to become complicated, and the diversity index will also show an upward trend; thus, the ability of forests to resist external disturbances will become stronger. Furthermore, this study found that the species, individuals, and diversity of insect species in the subalpine meadow ecosystem at an altitude of 2800 m were significantly different from those in the forest ecosystem ranging from 1600 m to 2600 m.

In this study, we found significant differences in the physical and chemical properties of soil at different altitudes, and the variation trend of each soil factor was different along the elevation gradient. Numerous studies have demonstrated that the physical and chemical properties of soil can affect the structure and diversity of insect communities by changing the growth of above-ground vegetation communities in forest ecosystems [17,18,19,20]. Therefore, prior to this investigation, we speculated that the physical and chemical properties of soil in forest ecosystems may have significant effects on the structure, distribution patterns, and diversity of insect communities. The results of RDA and correlation analysis also support the above speculation, and indicate that soil physicochemical properties, such as soil temperature, RH, pH, EC, AN, AP, AK, and BD, were closely related to the distribution and diversity of insect taxa orders along the altitude gradient.

Previous studies have confirmed that the spatial and temporal distribution of insect community composition can be influenced by abiotic and biotic factors, including plant communities, soil properties, and elevation gradients [6,7,8,9,10,11,12]. As the most important abiotic factors affecting insect richness and population structure distribution [61], the values of soil temperature and humidity in the forest ecosystems showed a significant downward trend with increasing altitude gradient (1600–2600 m). In addition, the soil temperature and relative humidity of the subalpine meadow at 2800 m were slightly higher than those of the forest ecosystem at lower altitudes (1600–2600 m), which may be caused by differences in surface vegetation type [9,53,54,55]. As an important insect community in the forest ecosystem at an altitude of 1600 m, the distribution of the Diptera and Coleoptera communities was positively correlated with soil temperature and negatively correlated with soil RH. Moreover, RDA showed that soil temperature was the main environmental variable affecting the distribution of insect groups along the altitude gradient, which was specifically manifested as a significant influence on the distribution of Diptera and Thysanoptera communities and was positively correlated with the number of insect species, individuals, and Pielou evenness index.

In this study, soil pH was found to be an important environmental factor that negatively affected the distribution pattern and diversity of insect communities along the elevation gradient. In addition, the soil water-related physical properties changed significantly with increasing elevation gradient, in which the soil bulk density and non-capillary porosity showed an increasing trend, whereas the other indices showed an obvious decreasing trend. However, these soil properties had marginal effect on the distribution patterns and diversity of the insects [62]. In contrast, soil fertility seems to have a significant impact on the distribution patterns and diversity of insect communities along the elevation gradient. The results showed that the soil nutrients AN, AP, and AK showed a downward trend with an increase in altitude gradient, but the ratios of N/P, N/K, and P/K were relatively stable. A possible reason for this is that the higher soil nutrient content in forest ecosystems can promote the growth of plants to a certain extent, thus increasing the species and number of insect populations [17,18,19,20]. However, the relationship between soil fertility and plant community species has not been further investigated in this study and requires further analysis in subsequent studies.

## 5. Conclusions

In this study, the distribution pattern and diversity of insect communities along altitude gradients of 1600–2800 m were analyzed in Guandi Mountain, China. The results showed that the insect community showed certain differentiation characteristics with the altitude gradient. The results of RDA and correlation analysis indicated that soil physicochemical properties, such as soil temperature, RH, pH, EC, AN, AP, AK, and BD, were closely related to the distribution and diversity of insect taxa orders along the altitude gradient. In addition, the soil temperature showed an obvious decreasing trend with increasing altitude, and temperature was also the most significant environmental factor affecting the insect community structure and diversity on the altitude gradient. Our results suggest that the interactive effects of altitude and environmental variables play an important role in determining the community structure, distribution patterns, and diversity of insect populations.

## Figures and Tables

**Figure 1 insects-14-00224-f001:**
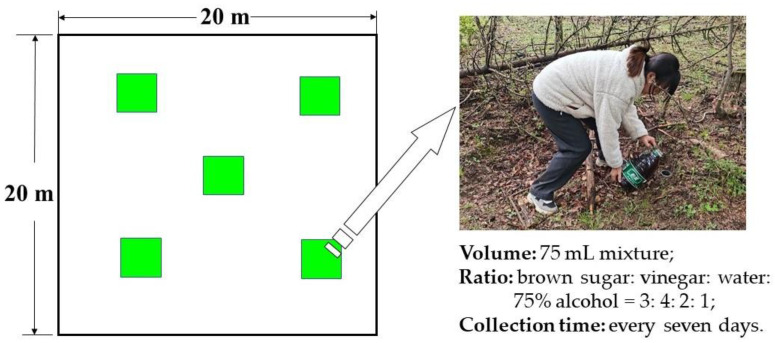
The schema of “pitfall trapping” sampling design.

**Figure 2 insects-14-00224-f002:**
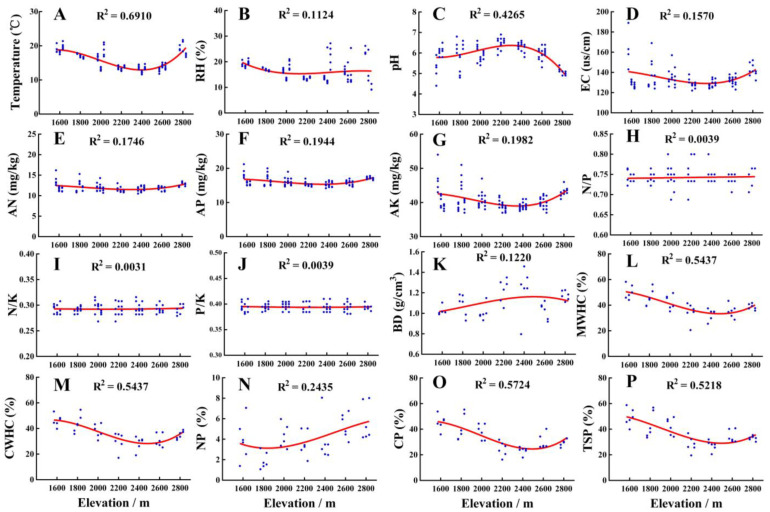
Variations of soil physicochemical properties along altitudinal gradient in the Guandi Mountain. (**A**): soil temperature; (**B**): soil relative humidity; (**C**): soil pH; (**D**): soil electric conductivity; (**E**): soil available nitrogen; (**F**): soil available phosphorus; (**G**): soil available potassium; (**H**): the ratio of available nitrogen to available phosphorus; (**I**): the ratio of available nitrogen to available potassium; (**J**): the ratio of available phosphorus to available potassium; (**K**): soil bulk density; (**L**): soil maximum water holding capacity; (**M**): soil capillary water holding capacity; (**N**): soil noncapillary porosity; (**O**): soil capillary porosity; (**P**): total soil porosity.

**Figure 3 insects-14-00224-f003:**
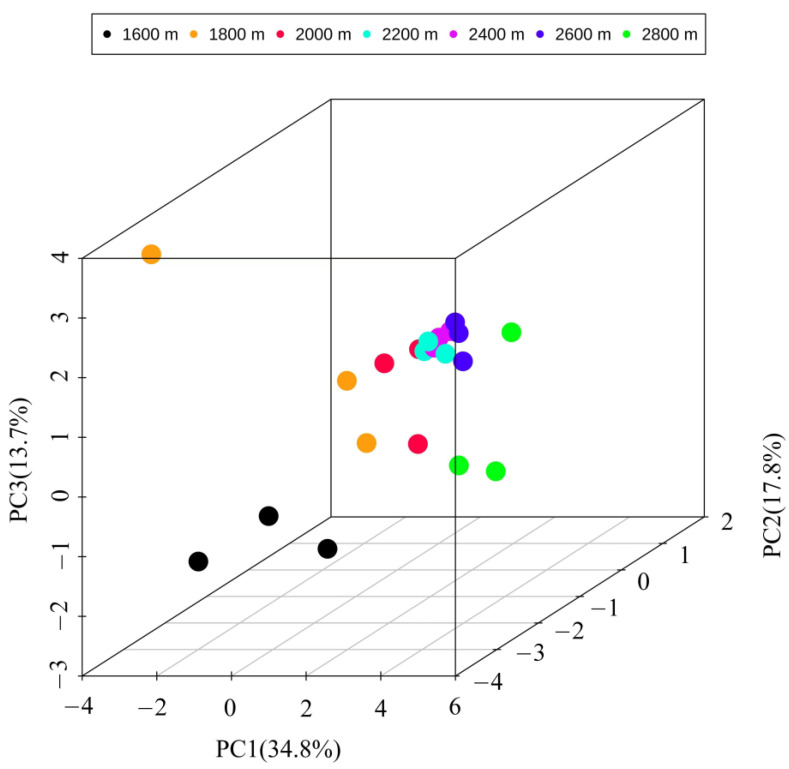
Principal components analysis of insect community along altitudinal gradient in the Guandi Mountain.

**Figure 4 insects-14-00224-f004:**
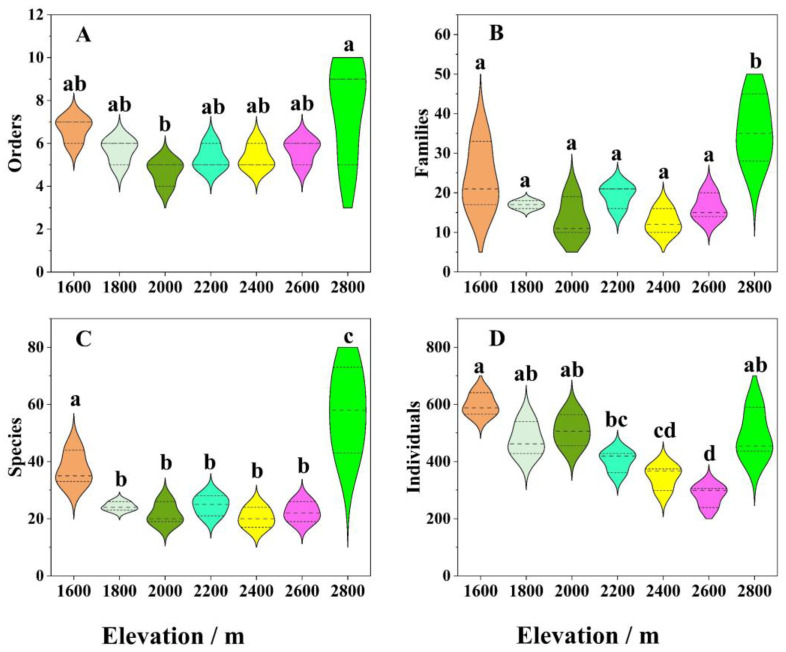
The difference in insect composition in typical vegetation community ecosystems along altitudinal gradient of the Guandi Mountain. (**A**): Orders; (**B**): Families; (**C**): Species; (**D**): Individuals. Note: Values are mean ± SD of three replicates for each typical vegetation community ecosystem. For each column, values with different letters are significantly different at *p* = 0.05.

**Figure 5 insects-14-00224-f005:**
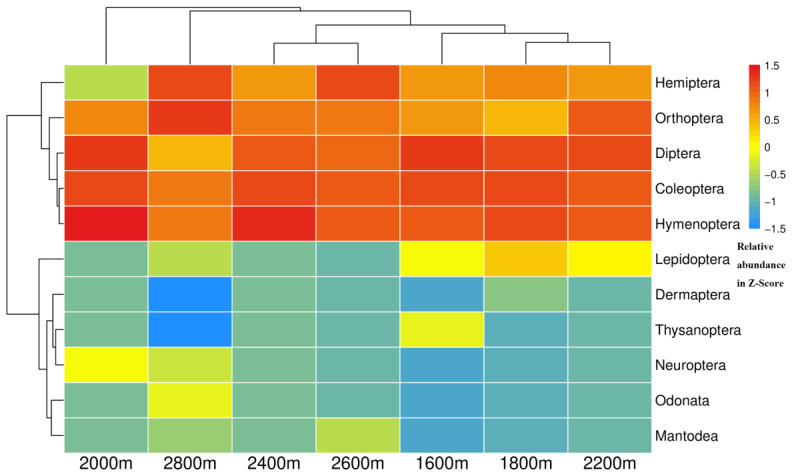
Heatmap of the insect taxa order along the altitudinal gradient in the Guandi Mountain.

**Figure 6 insects-14-00224-f006:**
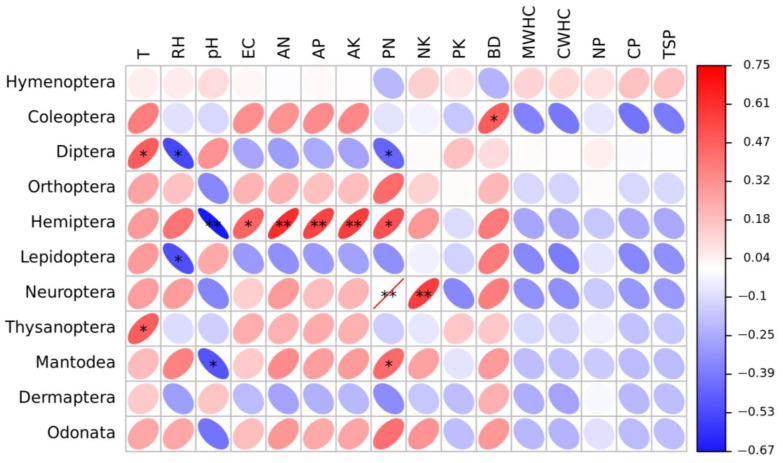
Pearson correlation coefficients between insect taxa order and soil factors in the Guandi Mountain. Left column is the taxa of insect order investigated in this study. Top row is the physical and chemical properties of the soil, where T means soil temperature, RH means soil relative humidity, pH means soil pH, EC means soil electric conductivity, AN means soil available nitrogen, AP means soil available phosphorus, AK means soil available potassium, PN means the ratio of available phosphorus to available nitrogen, NK means the ratio of available nitrogen to available potassium, PK means the ratio of available phosphorus to available potassium, BD means soil bulk density, MWHC means soil maximum water holding capacity, CWHC means soil capillary water holding capacity, NP means soil noncapillary porosity, CP means soil capillary porosity, TSP means total soil porosity. The values in the scale bar represent the correlation coefficients. The color of the circle indicates the direction of the correlation, where red indicates positive correlation and blue indicates negative correlation, and the darker the color indicates stronger correlation. The size of the circle indicates the strength of the correlation, and the larger the circle, the stronger the correlation. * *p* < 0.05, ** *p* < 0.01.

**Figure 7 insects-14-00224-f007:**
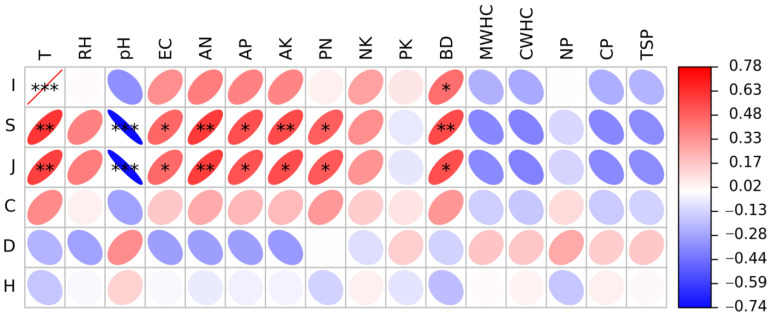
Pearson correlation coefficients between insect diversity and soil factors in the Guandi Mountain. Left column is the insect diversity index, where *I* means the numbers of insect individuals, *S* means the numbers of insect species, *J* means Pielou evenness index, *C* means Simpson index, *D* means Margalef richness index, *H* means Shannon–Wiener index. Top row is the physical and chemical properties of the soil, where T means soil temperature, RH means soil relative humidity, pH means soil pH, EC means soil electric conductivity, AN means soil available nitrogen, AP means soil available phosphorus, AK means soil available potassium, PN means the ratio of available phosphorus to available nitrogen, NK means the ratio of available nitrogen to available potassium, PK means the ratio of available phosphorus to available potassium, BD means soil bulk density, MWHC means soil maximum water holding capacity, CWHC means soil capillary water holding capacity, NP means soil noncapillary porosity, CP means soil capillary porosity, TSP means total soil porosity. The values in the scale bar represents the correlation coefficients. The color of the circle indicates the direction of the correlation, where red indicates positive correlation and blue indicates negative correlation, and the darker the color indicates stronger correlation. The size of the circle indicates the strength of the correlation, and the larger the circle, the stronger the correlation. * *p* < 0.05, ** *p* < 0.01, *** *p* < 0.001.

**Figure 8 insects-14-00224-f008:**
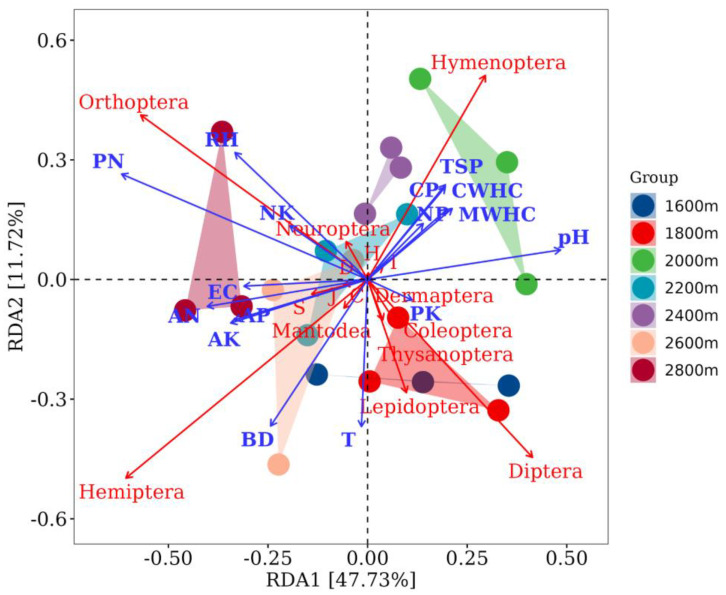
Redundancy analysis of the insect taxa order with soil factors in the Guandi Mountain. The red letters indicate the insect orders investigated in this study. The blue letters indicate the physical and chemical properties of the soil, where T means soil temperature, RH means soil relative humidity, pH means soil pH, EC means soil electric conductivity, AN means soil available nitrogen, AP means soil available phosphorus, AK means soil available potassium, PN means the ratio of available phosphorus to available nitrogen, NK means the ratio of available nitrogen to available potassium, PK means the ratio of available phosphorus to available potassium, BD means soil bulk density, MWHC means soil maximum water holding capacity, CWHC means soil capillary water holding capacity, NP means soil noncapillary porosity, CP means soil capillary porosity, TSP means total soil porosity.

**Table 1 insects-14-00224-t001:** Stand characteristics of seven typical vegetation community ecosystems along the altitude gradient in the Guandi Mountain.

Vegetation Community ^a^	Elevation (m)	Slope (°)	Plant Coverage (%)	Number of Individuals (Tree ha^−1^)	Mean DBH ^b^ (cm)	Mean Tree Height (m)
QWF	1600	26.16 ± 1.41	82.67 ± 2.52	1058.33 ± 101.04	9.36 ± 2.00	9.12 ± 2.55
PTF	1800	21.25 ± 2.02	81.67 ± 10.12	741.67 ± 128.29	17.44 ± 0.65	13.67 ± 2.43
PBM	2000	12.93 ± 5.20	73.50 ± 2.12	516.67 ± 14.43	17.15 ± 3.57	12.62 ± 1.55
PWF	2200	18.62 ± 6.25	82.33 ± 4.04	816.67 ± 80.36	16.98 ± 2.73	14.89 ± 1.80
PLF	2400	23.33 ± 4.16	77.67 ± 2.08	483.33 ± 38.19	30.82 ± 4.95	18.31 ± 3
LPF	2600	26.36 ± 6.53	76.33 ± 1.53	491.67 ± 87.80	25.38 ± 2.46	16.54 ± 2.78
SAM	2800	17.16 ± 3.28	92.67 ± 1.56			

^a^ QWF—Quercus wutaishansea forest; PTF—Pinus tabulaeformis forest; PBM—Populus davidiana and Betula platyphylla mixed forest; PWF—Picea wilsonii forest; PLF—P. wilsonii and Larix principis-rupprechtii mixed forest; LPF—L. principis-rupprechtii forest; SAM—subalpine meadow. ^b^ DBH—diameter at breast height of live trees.

**Table 2 insects-14-00224-t002:** Composition of insect communities in typical vegetation community ecosystems along altitudinal gradient of the Guandi Mountain.

Order	Family		Species		Individual	
	Number	Percentage/%	Number	Percentage/%	Number	Percentage/%
Coleoptera	23	28.75	73	33.03	2118	22.72
Diptera	18	22.50	35	15.84	2068	22.19
Orthoptera	11	13.75	35	15.84	1214	13.02
Hemiptera	11	13.75	33	14.93	1064	11.42
Hymenoptera	7	8.75	21	9.50	2656	28.49
Lepidoptera	4	5.00	18	8.14	109	1.17
Neuroptera	2	2.50	2	0.90	22	0.24
Dermaptera	1	1.25	1	0.45	16	0.17
Odonata	1	1.25	1	0.45	6	0.06
Mantodea	1	1.25	1	0.45	9	0.10
Thysanoptera	1	1.25	1	0.45	39	0.42
Total	80	100	221	100	9321	100

**Table 3 insects-14-00224-t003:** The Hill numbers of insect species composition and diversity in typical vegetation community ecosystems along altitudinal gradient of the Guandi Mountain.

Elevation (m)	Species Richness	Shannon–Wiener Index	Inverse Simpson’s Index	Berger–Parker Index
1600	38.00 ± 7.00 ^a^	3.26 ± 0.21 ^a^	0.05 ± 0.01 ^a^	0.12 ± 0.04 ^a^
1800	24.33 ± 1.53 ^b^	2.66 ± 0.45 ^b^	0.11 ± 0.07 ^ab^	0.19 ± 0.09 ^ab^
2000	22.67 ± 3.06 ^b^	2.41 ± 0.10 ^b^	0.15 ± 0.03 ^b^	0.31 ± 0.08 ^b^
2200	25.00 ± 4.00 ^b^	2.76 ± 0.47 ^ab^	0.11 ± 0.08 ^ab^	0.25 ± 0.15 ^ab^
2400	22.00 ± 4.36 ^b^	2.51 ± 0.29 ^ab^	0.13 ± 0.06 ^ab^	0.28 ± 0.12 ^ab^
2600	22.33 ± 3.51 ^b^	2.77 ± 0.35 ^ab^	0.08 ± 0.04 ^ab^	0.16 ± 0.05 ^ab^
2800	60.67 ± 14.19 ^c^	3.12 ± 0.31 ^ab^	0.08 ± 0.03 ^ab^	0.18 ± 0.05 ^ab^
*p*-value	0.00	0.07	0.29	0.21

Note: Values are mean ± SD of three replicates for each typical vegetation community ecosystem. For each row, values with different letters are significantly different at *p* = 0.05.

## Data Availability

The data presented in the study are available in the article.

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
