# Peer review of "Effects of Environmental Factors on the Spatial Distribution Pattern and Diversity of Insect Communities along Altitude Gradients in Guandi Mountain, China"

_insects, 2023, doi:10.3390/insects14030224_

Round 1

Reviewer 1 Report

This was a well written paper on an interesting topic. I thought your analyses were well done and the conclusions were supported by the results. I mainly have comments on the clarity of a couple items in the paper. First, the terms 'Guandi Mountain' and 'Guandi Mountains' were used almost interchangeably. I think there needs to be distinction between whether you focused on a single peak named Guandi or if you conducted your study in a mountain range known as the Guandi Mountains. Once established, use consistent terminology throughout the paper.

Along these lines, it was unclear if your study sites were on a single mountain or on several mountains with the plant communities of interest. I suggest providing the coordinates for all 21 study sites by adding a column for them in Table 1. 

Third, your figures were jumbled, possibly in the process of converting your paper into a pdf when submitting the manuscript. I still was able to sus out roughly which figures went with the different captions. However, some captions and figures need further explanation. I have provided suggestions on these in the body of the attached commented manuscript.

Author Response

Thanks so much for reviewer 1 for his/her positive comments and clear advices, which help improve the quality of the manuscript. We have studied the comments carefully and have revised and corrected the manuscript as reviewer mentioned. Revised portions are marked in yellow green in the revised manuscript. The point to point responses are as following:

Point 1: The interchangeable use of 'Mountains' (plural) and 'Mountain' (singular) throughout the paper is confusing. I suggest an explanation. For example, is there one peak or mountain known as 'Guandi Mountain' within a range of mountains referred to as the 'Guandi Mountains'? If so, does this study focus primarily on the individual mountain or on the range. If it focuses on the range instead of the individual mountain, then I suggest only referring to the individual mountain when sampling or analysis specifically involves that specific peak. Otherwise, use the term 'Mountains' throughout the abstract and the paper.

Response 1: Thanks very much for reviewer’s clear advice. We are very sorry for the confusion about 'Mountains' (plural) and 'Mountain' (singular) throughout the paper. In fact, the study area of this study refers to the Guandi Mountain, which located in the middle range of the Lvliang Mountains. Therefore, we went through the MS carefully and the word “Mountain” has been used throughout the abstract and the paper (L26, L80, L99, L101, L108, L116, L124, L197-198, L232, L236, L238, L240, L241, L243, L266-267, L280-281, L314, L362, and L441).

Point 2 (L11 and L91) : Both here and in the Short Summary the term 'urgent need' is used. Other than a lack of knowledge of diversity in this area/on this peak, what makes it urgent? Is there is an eminent change (mining, dam construction, building development, etc.)? If so, then that would need to be added to the introduction to support the 'urgent' statement. I agree this knowledge is important, so I suggest phrasing it differently. Something like: To better understand the impact environmental factors exert on insect population structure and diversity patterns in this area, we initiated a study.

Response 2: Thanks very much for reviewer’s useful advices. Considering reviewer’s suggestion, the term “urgent need” has been rewrote into “it is necessary” in L11 and L91.

Point 3 (L108) : Should this be 'brown'?

Response 3: We are sorry that this is a mistake in the spelling of the word, thanks very much for reviewer’s useful advices. The word “brow” has been revised to “brown” in L112.

Point 4: Was this study conducted on a single mountain with three sites at each vegetation type along an upward-sloping transect? Or, were the different vegetation types identified on an array of slopes/mountains at the desired vegetation? If multiple mountains were used for the different study plots, the coordinates for each study plot need to be included; I suggest including the coordinates in Table 1 by adding a column.

Response 4: Thanks very much to the reviewer for your clear suggestions. This study was investigated on the Guandi Mountain, which located in the Luliang Mountain range and with three sites for each vegetation type along an upward sloping transect. Combined with the reviewers' comments, we decided to add the geographical coordinates for all 21 study sites in the supplementary materials of the MS (Table S1). Moreover, the sentence “Table S1: The geographical coordinates of seven typical vegetation community ecosystems along the altitude gradient in the Guandi Mountain” has been added in the Supplementary Materials of the MS (L437-438).

Point 5: Is this the same as 'sugar-acetic acid-ethanol-water solution trap' method? If so, only use one term for the method. I suggest changing the term for this method to 'pitfall trapping'. the 'sugar-acetic acid......' and 'five-point sampling' are just describing what you put in the traps and how you arranged them at each site. 'Pitfall trapping' is a more concise and descriptive term. You already describe the solution and the placement method, so just change the term to which they refer.

Response 5: According to reviewer’s advice, the “sugar-acetic acid-ethanol-water solution trap” in L130 and “five-point sampling” in L135 has been changed into “pitfall trapping”.

Point 6: Using 'five-point...' here and above further justifies changing the sampling terminology. Again, using this term for this particular method only describes how sampling was arranged in the site. For this method, I suggest using the term 'soil core sampling'. This term will distinguish this method from the other method and clarify things for the reader.

Response 6: Thanks very much for reviewer’s helpful advices, the phrase of “five-point sampling method” has been replaced by “soil core sampling” in L146.

Point 7: There is no need to have 'Figure 1' attached to all the subfigure letters in the figure caption. Just have the letters representing the different graphs.

Response 7: According to reviewer’s advice, the “Figure 1” has been deleted in the Figure 1 caption in L197-203.

Point 8: Something rearranged your table and figure placement when you submitted your manuscript. Make sure they are reordered in the correct place.

Response 8: Thank you very much for the reviewer's reminder, we went through the MS and put all the figures and tables in the right place.

Point 9 (Figure 2) : I don't understand what this figure is showing. It seems to show the correlation of communities at one elevation with other communities. However, I'm unsure what this illustrates, other than some are different from others. I think one of the axes needs to be relabeled and a more descriptive caption included (both show elevation class) in order for interpretation to be more intuitive. Or, perhaps eliminate it, as you present more detailed correlation heat maps later in the paper.

Response 9: Considering reviewer’s suggestion, Figure 2 (Pearson correlation analysis of insect community along the altitudinal gradient in the Guandi Mountain) has been removed from the MS.

Point 10: For Figures 2 (if you keep it), 5, 7, and 8 a legend needs to accompany the rainbow scale at the right of each heatmap. For Figures 2, 7 and 8, it is pseudo-intuitive that the scale represents correlation coefficients, but it still needs a label. However, what does the scale stand for in Figure 5, as the values exceed 1 (and -1). For all figures, do not assume the reader is as familiar with your data and results as you are. Symbology in all figures needs to be explained well.

Response 10: Thanks very much to the reviewer for your clear suggestions. According to reviewer’s advice, the legend “Relative abundance in Z-Score” for Figure 5 (and now Figure 4) has been added under the rainbow scale. Moreover, the sentence “The values in the scale bar represents the correlation coefficients” have been added into the captions of Figure 7 and Figure 8, respectively.

Point 11: Graphic needs to be higher resolution so it can be more clearly examined. All font sizes in the graphic should be increased for ease of readability.

Response 11: Thanks very much for reviewer’s useful advices, the resolution of all the Figures has been improved and the font sizes has been increased in the MS.

Point 12 (Figure 7 and Figure 8): This (and Figure 8) is a good figure, but it needs more explanation. Label the scale bar on the right of the graph, what do the direction of the ovals indicate? What do the size of the ovals indicate? What p-levels to the asterisks indicate? These thing should be in your figure captions.

Response 12: According to reviewer’s advice, the sentences of “The values in the scale bar represents the correlation coefficients. The color of the circle indicates the direction of the correlation, where red indicates positive correlation and blue indicates negative correlation, and the darker the color indicates stronger correlation. The size of the circle indicates the strength of the correlation, and the larger the circle indicates the stronger the correlation. * P<0.05, ** P<0.01, ***P<0.001” have been added into the captions of Figure 7 and Figure 8, respectively.

Response 13: Moreover, in order to further improve the quality of the MS, we have gone through the manuscript again to check the possible problems and made corresponding modifications, as detailed below:

  1. The word “latitude” has been deleted in L108.
  2. Due to the addition and reduction of some references, the order of references in MS has been adjusted to some extent, and we have also marked them in MS accordingly.

Reviewer 2 Report

You have an interesting contribution, yet basic terminology, concepts, and premises should be carefully revised in order to have a coherent piece of work, including concise research goals.  Please go over my comments to the text in the attached file, there you will have several hints to polish your work (of course these are all suggestions at the end).  It should be clear from the onset (I would suggest from the title), that you are emphasizing soil fauna and soil environmental variables, although you also sampled tree or canopy fauna, this should also be clarified in the introduction and probably in the methods.  There seems to be inconsistency (or perhaps use of a little too many...) in concepts/terms, such as "insect community structure and composition", "alpha diversity", "spatial distribution", "composition and spatial distribution patterns", "underlying maintenance mechanisms of insect species", "populations", please increase consistency and clarity.

Author Response

Thanks so much for reviewer 2 for his/her positive comments on our work. We have revised the manuscript as reviewer mentioned. Those comments are all valuable and very helpful for revising and improving our manuscript, as well as the important guiding significance to our researches. We have studied the comments carefully and have revised and corrected the manuscript which we hope meet his/her approval. Revised portions are marked in yellow in the revised manuscript. The responses are as following:

Point 1 (L11): construction?

Response 1: Thanks very much for reviewer’s useful advice. The word “construction” has been replaced by the word “structure” in L11.

Point 2 (L11) : urgent need?

Response 2: Thanks very much for reviewer’s useful advices, perhaps this expression is not appropriate in MS. Considering reviewer’s suggestion, the term “urgent need” has been rewrote into “it is necessary” in L11 and L91.

Point 3 (L16) : Please define RDA.

Response 3: Many thanks to the reviewers for their good suggestions. According to the comments of the reviewers, the definition of RDA “redundancy analysis (RDA)” has been added in L16.

Point 4 (L29) : delete analysis.

Response 4: Thanks very much for reviewer’s clear advices. According to reviewer’s advice, the ”analysis” has been deleted from the MS.

Point 5 (L41) : substance?

Response 5: Thanks very much for reviewer’s useful advice. The word “substance” has been replaced by the word “nutrients” in L11.

Point 6 (L42-43) : A vague statement, try to be more specific, use better terms, for instance “insects contain a great number of biological resources”, what does it mean?  insects help maintain?  Help utilize?  Represent an important biological resource?  The ocean, probably, contains a great number of biological resources…

Response 6: Thanks very much for reviewer’s clear advice. As we all know, insects are the largest population and the largest number of individuals on the earth. At the same time, insects can be divided into different groups, such as medicinal insects, natural enemies, and saprophagous insects, which play huge biological roles in different ecosystems. Therefore, we can consider insects as a biological resource. Moreover, to avoid causing ambiguity, we decided to delete "and contain a great number of biological resources" from the MS.

Point 7 (L46-47) : Isn’t this somewhat contradictory?

Response 7: Thanks very much for reviewer’s clear advice. To avoid causing ambiguity, we decided to delete " and can indicate environmental changes to a certain extent" from the MS.

Point 8 (L68-69) : Is there internal climate?

Response 8: Thanks very much for reviewer’s useful advices. What we want to express here is that insects, as typical ectotherms, are greatly affected by changes in the temperature of the external environment. There is no doubt that insects are also affected by internal climate factors.

Therefore, to avoid causing ambiguity, we decided to delete "external climate" from the MS.

Point 9 (L75-79) : delete L71-79.

Response 9: According to reviewer’s advice, the sentence “Therefore, understanding the vertical spatial patterns of insects may not only improve our understanding of the structure and dynamics of insect communities, but also of the patterns of variation in biodiversity in response to environmental factors along latitudinal gradients, as well as the effects of global climate change on insect communities” in L71-79 has been deleted from the MS.

Point 10 (L93) : above six types of vegetation were described, please double check for consistency and clarity, same with table 1…

Response 10: Thanks very much for reviewer’s helpful advices. According to the comments of the reviewers, a detailed description of the seven different ecosystems was placed in the introduction section. The details can be viewed in L88-92. Combined with the reviewers' comments, we decided to add the geographical coordinates for all 21 study sites in the supplementary materials of the MS (Table S1). Moreover, the sentence “Table S1: The geographical coordinates of seven typical vegetation community ecosystems along the altitude gradient in the Guandi Mountain” has been added in the Supplementary Materials of the MS (L432-433).

Point 11 (L133) : could it be more specific..? July through August? Two months? n days?

Response 11: Thanks very much for reviewer’s useful advices. Considering the suggestions of the reviewer, the frequency and time of the insect specimens collection had been written in L133-134 Insect specimens were collected every seven days, and the survey time was concentrated from early July to the end of August in 2020 and 2021” in the MS.

Point 12 (L134): specimens?

Response 12: Thanks to the reviewer's reminder, “specimens” was incorrectly written as “species” in L134. We have made modifications in MS.

Point 13 (L141): for preservation? to be preserved…?

Response 13: Thanks very much for reviewer’s helpful advices. According to reviewer’s advice, the phrase “to preserve” has been replaced by “for preservation” in the MS. Additionally, the phrase of “the insect specimens” has been removed from the MS.

Point 14 (Figure 7 and Figure 8): shape and orientation of ellipses mean something? Asterisks?

Response 14: Many thanks to the reviewer for his/her useful suggestions. According to reviewer’s advice, the sentences of “The values in the scale bar represents the correlation coefficients. The color of the circle indicates the direction of the correlation, where red indicates positive correlation and blue indicates negative correlation, and the darker the color indicates stronger correlation. The size of the circle indicates the strength of the correlation, and the larger the circle indicates the stronger the correlation. * P<0.05, ** P<0.01, ***P<0.001” have been added into the captions of Figure 7 and Figure 8, respectively.

Point 15 (L337) : insect population?

Response 15: Thanks very much for reviewer’s clear advices. The phrase “insect population” has been replaced by “number of insects” in L334.

Point 16 (L346) : careful with the use of the term or concept of population, here you probably mean species… (?)

Response 16: Thanks very much for reviewer’s clear advices. According to reviewer’s advice, the word “population” has been replaced by “species” in L334.

Point 17 (L355-357) : hard to grasp from the results, perhaps a figure illustrating this would help…

Response 17: Thanks very much for reviewer’s clear advices. In fact, Figure 4 “Heatmap of the insect taxa order along the altitudinal gradient in the Guandi Mountain” can explain this phenomenon.

Point 18 (L405-407) : Not quite evident from results the way are depicted now…, at least not apparent from figure legends… (!)

Response 18: Thanks very much for reviewer’s clear advices. Figure 7 shows the pearson correlation coefficients between insect diversity and soil factors in the Guandi Mountain. From this figure, we can clear see that both the number of insect species (S) and Pielou evenness index (J) were significantly negatively correlated with soil pH (P<0.001). In addition, soil pH also shows a negative relationship with the numbers of insect individual (I) and Simpson index (C).

Response 19: Moreover, in order to further improve the quality of the MS, we have gone through the manuscript again to check the possible problems and made corresponding modifications, as detailed below:

  1. The word “latitude” has been removed from L108.
  2. Due to the addition and reduction of some references, the order of references in MS has been adjusted to some extent, and we have also marked light blue in MS accordingly.
  3. Figure 2 (Pearson correlation analysis of insect community along the altitudinal gradient in the Guandi Mountain) has been removed from the MS.

Reviewer 3 Report

This paper investigated the Effects of Environmental Factors on the Spatial Distribution Pattern and Diversity of Insect Communities along Altitude Gradients in Guandi Mountain, China

As the topic and the generated dataset seem interesting, I found some inconsistencies in the relevance of the used references, and the used statistical methods. I suggest to the authors to revise the relevance of all the reference and change if this sound not relevant for the statement where there are used.

As it is a deep sampling of all insect, a better explanation on the taxonomic aspect must be provided. The paper present just order taxa but we do not have a list of the insect family and species which generate irrelevant analysis on species diversity. So, this need to be improve.

Many results are not methodologically introduce in the M&M, please re-check this. In the discussion part, a furhter interpretation of the main families or insect would be welcome as you want to submit your work in Insects. Maybe when you will send to us the revised document the conclusion may change. 

L43 : Can you explain what you mean by "biological resources" ? And add this precision in the MS

L44: I think that insect also have in other ecosystem

L52: Reference 6,7,9 are not relevant for your statement please find more relevant reference or rearrange the sentence

L55: Same remark here

L58-60: Please place reference after their statement; reference 16 is more appropriate for plant growth than for insect communities

L83-89: Please add the first descriptor of each plant species as it is mandate by the classical taxonomy rules

L93: Which seven different ecosystem I think this is the different forest vegetation so please precise this? If they are not introduce in the introduction part, I suggest to introduce the different ecosystem by one paragraph in the introduction part

L106-107 : Reference of the climate ecosystem region ?

L112-116: This is redundant with the introduction. Find a way to associate both in just one part.

L128: Write it in the M&M text

L130-132 : Can you give a picture of the “sugar–acetic acid–ethanol–water solution trap” ? 

L134-135 : I do not think that the number of net swap is important, a duration would be more appropriate. As  netting technique is more dependent on the user experience.

L141-144: Do you prepare the collected specimens and vouchered it in insect collection ? Which references do you used to identify the collected, which taxonomic key ? Do you also use insect collection reference ? If yes please mention it.

L129 : Finally, what is the frequency of the sampling collection ? How many times do you collect insects at each locations ? This need to be mentioned. A schema of the sampling design could be interesting to better understand your sampling design.

L166-170: This paper presents an interesting study, but the use of Alpha Diversity Indices presents serious conceptual and statistical problems, which make comparisons of species richness or species abundances across communities nearly impossible. I present below the best solution for solve the problem.

The problems: Both Shannon and Simpson Indices are the sum of calculations (number of species) and consequently are biased by sampling effort. I do not know about Pielou or Margalef, but this seems on the same biaeses.

The solution: The best way forward is the application of Hill Numbers (see Jost, 2006; and Tuomisto, 2010), that provide a framework to integrate the previous concepts into a consistent terminology, based on the paper by Hill (1973). The Hill numbers, also called the “effective number of species”, as the best choice to quantify abundance-based species diversity. Here, the effective number of species means the number of equally abundant species that are needed to give the same value of a diversity measure. Therefore, the best solution is to use the sequence o Hill Numbers as follows: N0 = species richness (S); N1= Exponential Shannon-Wiener (exp H´); N2 Inverse Simpson´s index (1/D); N4 = Berger-Parker index (1/d)(see also Magurran 2004).

  References

Hill M. O. 1973. Diversity and evenness: a unifying notation and its consequences. Ecology 54, 427–43210.2307/1934352 (doi:10.2307/1934352)

Jost L. 2006. Entropy and diversity. Oikos 113, 363–37510.1111/j.2006.0030-1299.14714.x (doi:10.1111/j.2006.0030-1299.14714.x)

Magurran, A. E. 2004. Measuring Ecological Diversity. Blackwell Publishing, Oxford. 256 pp.

Tuomisto H. 2010. A consistent terminology for quantifying species diversity? Yes, it does exist. Oecologia 164, 853–86010.1007/s00442-010-1812-0 (doi:10.1007/s00442-010-1812-0)

Figure1: How did you generate the Figure 1 ? Please explain it in the M&M

Figure 9: Same here

L172-176: Instead of using correlation, you need to explore the Generalized Linear Mixed-Models (GLMM) to infer the effect of the environmental variables on the alpha diversity  of your insect communities, this will improve your conclusions of yours study. Correlations seems not very strong in that kind of research especially when you did not transform your data following theoritical distribution of errors. 

L213-214: I suggest that you introduce multivariate analysis in your M&M. Moreover, PCA could be replace by a PCoA which is more appropriate to analysis biodiversity data. See here https://link.springer.com/book/10.1007/978-3-319-71404-2

Moreover, species composition could be written after the dataset description. You also never explain RDA and why you use this, in the case of choosing PCoA, you must test db-RDA to match with the previous ordiantion analysis.

L220-222: Same remark here you did not introduce the cluster analysis and heatmap analysis in the M&M.

L227-231: After reading the Hill framework, I thin you will change this analysis, as this seems none relevant here.

Author Response

We are greatly appreciating for reviewer 3 for his/her clear advices and dedicated efforts, which help improve the quality of the manuscript. Those comments are all valuable and very helpful for revising and improving our manuscript, as well as the important guiding significance to our researches. We have studied the comments carefully and have revised and corrected the manuscript which we hope meet his/her approval. Revised portions are marked in light blue in the revised manuscript. The point to point responses are as following:

Point 1 (L43): Can you explain what you mean by "biological resources" ? And add this precision in the MS.

Response 1: Thanks very much for reviewer’s clear advice. As we all know, insects are the largest population and the largest number of individuals on the earth. At the same time, insects can be divided into different groups, such as medicinal insects, natural enemies, and saprophagous insects, which play huge biological roles in different ecosystems. Therefore, we can consider insects as a biological resource. Moreover, to avoid causing ambiguity, we decided to delete "and contain a great number of biological resources" from the MS.

Point 2 (L44) : I think that insect also have in other ecosystem.

Response 2: Thanks very much for reviewer’s clear advices. I agree with the reviewer that insects do exist in different ecosystems, such as farmland, forests, and cities. In fact, this study specifically refers to the function and role of insects in forest ecosystems. Considering reviewer’s suggestion, the term “forests” has been rewrote into “different ecosystems” in L43.

Point 3 (L52 and L55) : Reference 6,7,9 are not relevant for your statement please find more relevant reference or rearrange the sentence.

Response 3: Many thanks to the reviewers for their good suggestions. According to the comments of the reviewers, the more relevant references [6-12] were placed in the original sentences in L52. The details are as follows:

  1. Rasmann, S.; Pellissier, L.; Defossez, E.; Jactel, H.; Kunstler, G.; Bailey, J.K. Climate-driven change in plant-insect interactions along elevation gradients. Funct Ecol. 2014, 28(1), 46-54.
  2. Gao, R.; Shi, J.; Huang, R.; Wang, Z.; Luo, Y. Effects of pine wilt disease invasion on soil properties and Masson pine forest communities in the Three Gorges reservoir region, China. Ecol Evol. 2015, 5(8), 1702-1716.
  3. Hodkinson, I.D. Terrestrial insects along elevation gradients: species and community responses to altitude. Biol Rev. 2005, 80, 489–513.
  4. Moreira, X.; Abdala-Roberts, L.; Rasmann, S. Plant diversity effects on insect herbivores and their natural enemies: current thinking, recent findings, and future directions. Curr Opin Insect Sci. 2016, 14, 1-7.
  5. Cuevas-Reyes, P.; Quesada, M.; Hanson, P.; Dirzo, R.; Oyama, K. Diversity of gall-inducing insects in a Mexican tropical dry forest: the importance of plant species richness, life-forms, host plant age and plant density. J Ecol. 2004, 92, 707–716.
  6. Klingauf, F. Interrelations between Pests and Climatic Factors. In Food-Climate Interactions; Bach, W., Pankrath, J., Schneider, S.H., Eds.; Springer; Dordrecht, USA, 1981; pp. 285-301.
  7. Heinen, R.; Biere, A.; Harvey, JA.; Bezemer, T. Effects of Soil Organisms on Aboveground Plant-Insect Interactions in the Field: Patterns, Mechanisms and the Role of Methodology. Front Ecol Evol. 2018, 6,

Point 4 (L58-60) : Please place reference after their statement; reference 16 is more appropriate for plant growth than for insect communities.

Response 4: We apologize for misplacing the references and thank the reviewers for their helpful suggestions. We have readjusted the position of the references [12,17] and [18,20] in L59-60. The details are as follows:

  1. Heinen, R.; Biere, A.; Harvey, JA.; Bezemer, T. Effects of Soil Organisms on Aboveground Plant-Insect Interactions in the Field: Patterns, Mechanisms and the Role of Methodology. Front Ecol Evol. 2018, 6,
  2. Vergara, G.D.; Williams, L.G.; Casanoves, F. Leaf functional traits vary within and across tree species in tropical cloud forest on rock outcrop versus volcanic soil. J Veg Sci. 2019, 31(1), 129–138.
  3. Cisneros, J.J.; Godfrey, L.D. Midseason pest status of the cotton aphid (Homoptera: Aphididae) in California cotton: is nitrogen a key factor? Popul Ecol. 2001, 30(3), 501–510.
  4. Huberty, A.F.; Denno, R.F. Consequences of nitrogen and phosphorus limitation for the performance of two planthoppers with divergent life history strategies. Oecologia. 2006, 149(3), 444–455.

Point 5 (L83-89) : Please add the first descriptor of each plant species as it is mandate by the classical taxonomy rules.

Response 5: Thanks very much for reviewer’s clear advices. According to reviewer’s advice, the first descriptor of each plant species in L85-89 have been added in the MS.

Point 6 (L93) : Which seven different ecosystem I think this is the different forest vegetation so please precise this? If they are not introduce in the introduction part, I suggest to introduce the different ecosystem by one paragraph in the introduction part.

Response 6: Thanks very much for reviewer’s helpful advices. According to the comments of the reviewers, a detailed description of the seven different ecosystems was placed in the introduction section. The details can be viewed in L94-98.

Point 7 (L106-107) : Reference of the climate ecosystem region ?

Response 7: According to reviewer’s advice, the reference [31] of the climate ecosystem region has been added in L111. The details are as follows:

  1. Gao, J.; Zhang, Y. Distributional patterns of species diversity of main plant communities along altitudinal gradient in secondary forest region, Guandi Mountain, China. J Forestry Res. 2006, 17(2), 111-115.

Point 8 (L112-116) : This is redundant with the introduction. Find a way to associate both in just one part.

Response 8: Thanks very much for reviewer’s helpful advices. According to the comments of the reviewers, a detailed description of the seven different ecosystems was placed in the introduction section. The details can be viewed in L94-98 of MS.

Point 9 (L128) : Write it in the M&M text

Response 9: Considering reviewer’s suggestion, the phrase of “values are mean±SD” has been moved to the M&M text in L118.

Point 10 (L129) : Finally, what is the frequency of the sampling collection ? How many times do you collect insects at each locations? This need to be mentioned. A schema of the sampling design could be interesting to better understand your sampling design.

Response 10: According to reviewer’s advice, the frequency of the insect specimens collection had been written in L133-134 Insect specimens were collected every seven days, and the survey time was concentrated from July to August in 2020 and 2021” in the MS.

Point 11 (L130-132) : Can you give a picture of the “sugar–acetic acid–ethanol–water solution trap” ? 

Response 11: As suggested by another reviewer, the name of the “sugar–acetic acid–ethanol–water solution trap” was changed to "pitfall trapping" (L130). The method of “sugar–acetic acid–ethanol–water solution trap” is one of the most traditional methods of trapping insects in the field, and a detailed description of this method is available in L134-141. The picture of the "sugar -- acetic acid -- ethanol -- water solution trap" is as follows (Figure 1). If the reviewer feel it is necessary, we can put this picture in M&M or supplementary materials of the MS.

Figure 1  Field operation of "sugar -- acetic acid -- ethanol -- water solution trap".

Point 12 (L134-135) : I do not think that the number of net swap is important, a duration would be more appropriate. As netting technique is more dependent on the user experience.

Response 12: Thanks very much for reviewer’s useful advices. Considering the suggestions of the reviewer, the sentence “ For the “sweep net sampling”method, a diameter of 38 cm was used to collect insect species and the net was swept more than 200 times in each plot” has been revised to “The “sweep net sampling” method was used to collect low flying insect species, and the net was swept more than 200 times in each plot” in L133-134.

Point 13 (L141-144): Do you prepare the collected specimens and vouchered it in insect collection ? Which references do you used to identify the collected, which taxonomic key? Do you also use insect collection reference? If yes please mention it.

Response 13: Thanks to the reviewers for their questions. In this study, all insect specimens collected in the field were properly preserved and brought back to the laboratory for specimen preparation and identification. All the collected insects were identified mainly by morphological methods, and DNA barcoding was applied for molecular identification. Moreover, according to reviewer’s advice, references for insect identification has been inserted into L143-144. The references [34-41] cited in the MS for insect identification are as follows:

  • Yang, M.F .; Mang, Z.H.; Li, Z.Z. Fauna Sinica Insecta, Vol.67; Science Press, Beijing, China, 2017; pp. 637.
  • Fan, Z.D.; Deng, Y.H. Fauna Sinica Insecta, Vol.49; Science Press, Beijing, China, 2008; 1186.
  • He, J.H.; Chen, X.X.; Ma, Y. Fauna Sinica Insecta, Vol.18; Science Press, Beijing, China, 2000; 757.
  • Han, H.X.; Xue, D.Y. Fauna Sinica Insecta, Vol.54; Science Press, Beijing, China, 2011; 787.
  • Ren, G.D. et al. Fauna Sinica Insecta, Vol.63; Science Press, Beijing, China, 2016; 534.
  • Li, M.L. Resource Entomology. China ForestryPublishing House, Beijing, China, (in Chinese)
  • Chou, I. Monographia Rhopalocerorum Sinensium. Henan Scientific and Technological Publishing House, Zhengzhou, China, (in Chinese)
  • Hebert, P.; Cywinska, A.; Ball, SL.; Dewaard, JR. Biologicalidentification through DNA barcodes. Proc Biol Sci. 2003, 270(1512), 313‒321.

Point 14 (Figure 1 and Figure 9): How did you generate the Figure 1 and Figure 9? Please explain it in the M&M.

Response 14: Thanks very much for reviewer’s helpful advices. According to reviewer’s advice, the method of making Figure 1 L171: plotted using GraphPad Prism 6.0” and Figure 9L176-178: the redundancy analysis and mapping of environmental variables that affected insect community structure and diversity were performed using CANOCO 5.0” have been added in the MS.

Point 15 (L166-170): This paper presents an interesting study, but the use of Alpha Diversity Indices presents serious conceptual and statistical problems, which make comparisons of species richness or species abundances across communities nearly impossible. I present below the best solution for solve the problem. The problems: Both Shannon and Simpson Indices are the sum of calculations (number of species) and consequently are biased by sampling effort. I do not know about Pielou or Margalef, but this seems on the same biaeses. The solution: The best way forward is the application of Hill Numbers (see Jost, 2006; and Tuomisto, 2010), that provide a framework to integrate the previous concepts into a consistent terminology, based on the paper by Hill (1973). The Hill numbers, also called the “effective number of species”, as the best choice to quantify abundance-based species diversity. Here, the effective number of species means the number of equally abundant species that are needed to give the same value of a diversity measure. Therefore, the best solution is to use the sequence o Hill Numbers as follows: N0 = species richness (S); N1= Exponential Shannon-Wiener (exp H´); N2 Inverse Simpson´s index (1/D); N4 = Berger-Parker index (1/d)(see also Magurran 2004).

Response 15: Many thanks to the reviewer for his/her positive comments and useful suggestions. When writing this MS, we also refer to a large number of relevant literatures when conducting relevant data analysis. Alpha diversity index is a traditional concept that reflects the number of species and their relative abundance in a community and comparing the difference of alpha diversity index can show the difference of species diversity between different habitats. In addition, alpha diversity index analysis has a wide range of applications and recognition in the field of ecology. Therefore, I do not quite understand why this study has serious conceptual and statistical problems in using the alpha diversity index to analyze differences in insect species diversity in habitats at different altitudes, making cross-community species richness or species abundance comparisons nearly impossible. In order to avoid similar problems in the future, we hope the reviewer can give further instructions to better complete the follow-up research. Thank you very much.

Meanwhile, I would like to thank the reviewers for proposing to use Hill number to solve some problems caused by the analysis of alpha diversity index in order to optimize the quality of MS. According to the comments of the reviewers, we also calculated the Hill Number of the insect community changing with the elevation gradient in , as shown in Table 1 below. We temporarily put this table in the supplementary materials of the MS (Table S3). Moreover, the sentence “Table S3 The Hill numbers of insect species composition and diversity in typical vegetation community ecosystems along altitudinal gradient of the Guandi Mountain” has been added in the Supplementary Materials of the MS (L440-442). If the reviewer thinks that the following data analysis is more appropriate for this study than alpha diversity analysis, we will make further adjustments and analysis later. The relevant references are as follows.

Table 1 The Hill numbers of insect species composition and diversity in typical vegetation community ecosystems along altitudinal gradient of the Guandi Mountain

Elevation
(m)

Species richness

Shannon-Wiener index

Inverse Simpson´s index

Berger-Parker index

1600

38.00±7.00b

3.2571±0.2135a

0.0510±0.0120a

0.1185±0.0435b

1800

24.33±1.53c

2.6606±0.4485abc

0.1100±0.0669a

0.1937±0.0973a

2000

22.67±3.06c

2.4101±0.0957c

0.1478±0.0317a

0.3078±0.0812ab

2200

25.00±4.00c

2.7565±0.4696abc

0.1121±0.0790a

0.2520±0.1524ab

2400

22.00±4.36c

2.5097±0.2869bc

0.1349±0.0586a

0.2777±0.1161ab

2600

22.33±3.51c

2.7651±0.3534abc

0.0823±0.0362a

0.1614±0.0502ab

2800

60.67±14.19a

3.1178±0.3082ab

0.0800±0.0261a

0.1846±0.0465ab

References:

  1. Hill, M.O. Diversity and evenness: a unifying notation and its consequences. 1973, 54, 427-432
  2. Jost, L. Entropy and diversity. 2006, 113, 363-375.
  3. Magurran, A.E. Measuring Ecological Diversity. Blackwell Publishing, Oxford, UK, 2004; pp. 256.
  4. Tuomisto, H. A consistent terminology for quantifying species diversity? Yes, it does exist. 2010, 164, 853-860.

Point 16 (L213-214) : I suggest that you introduce multivariate analysis in your M&M.

Response 16: Thanks very much for reviewer’s clear advices. According to reviewer’s advice, the sentence “All environmental variables and insect metrics were compared using one-way analysis of variance (ANOVA) and Fisher’s least significant difference (LSD) tests with an alpha value of P < 0.05” has been added in the M&M.

Point 17 (L220-222): Same remark here you did not introduce the cluster analysis and heatmap analysis in the M&M.

Response 17: Thanks very much for reviewer’s helpful advices. According to reviewer’s advice, the sentences “Heatmaps (n = 21) for insect taxa order along the altitudinal gradient were plotted using heatmap tools on the Genescloud platform (https:// www. genes cloud. cn)” has been added in the M&M.

Response 18: Moreover, in order to further improve the quality of the MS, we have gone through the manuscript again to check the possible problems and made corresponding modifications, as detailed below:

  1. The word “latitude” has been removed from L108.
  2. Due to the addition and reduction of some references, the order of references in MS has been adjusted to some extent, and we have also marked light blue in MS accordingly.

Round 2

Reviewer 2 Report

Thanks for taking care of my observations, as well as of other reviewers, I believe the manuscript has considerably improved and is essentially ready for acceptance, perhaps after a last, quick, final polishing.  One minor thing that called my attention, the use of the expression "number of insects in the forest" (line 334 of the discussion), I would believe this is basically "abundance" (insect abundance values in the forest, for instance), as abundance is a proper term or concept of population biology (for your consideration).  Congratulations on an original piece of work with a novel approach.

Author Response

hanks so much for reviewer 2 for his/her positive comments and clear advices, which help improve the quality of the manuscript. We have studied the comments carefully and have revised and corrected the manuscript as reviewer mentioned. Revised portions are marked in yellow green in the revised manuscript. The point to point responses are as following:

Point 1: Thanks for taking care of my observations, as well as of other reviewers, I believe the manuscript has considerably improved and is essentially ready for acceptance, perhaps after a last, quick, final polishing. One minor thing that called my attention, the use of the expression "number of insects in the forest" (line 334 of the discussion), I would believe this is basically "abundance" (insect abundance values in the forest, for instance), as abundance is a proper term or concept of population biology (for your consideration). Congratulations on an original piece of work with a novel approach.

Response 1: Thanks very much for reviewer’s clear advice. At the same time, thank you very much for your review and recognition of our work. According to your suggestion, the word “number” has been changed to “abundance” in L336 of the MS.

Response 2: Moreover, in order to further improve the quality of the MS, we have gone through the manuscript again to check the possible problems and made corresponding modifications, as detailed below:

  1. Table S3 “List of the collected insects in the Guandi Mountain” has been added in the supplementary materials of the MS.
  2. “Table S1”, “Table S2” and “Table S3” have been added in L111, L179, and L204 respectively.
  3. Due to the addition and reduction of some references, the order of references in MS has been adjusted to some extent, and we have also marked light blue in MS accordingly.
  4. We have removed the original Figure 5 from MS, therefore, the order of Figures in MS has been adjusted to some extent, and we have also marked light blue in MS accordingly.

Reviewer 3 Report

Thanks for answers.

Response 7. Please note the original reference for the climate zone and not from forest paper. This mean that you can cite reference(s) from public climate data (maybe from china). 

Response 10. The schema of the sampling design is missing... in the manuscript. 

Response 11. Yes in the supplemental information, the figure could be added and referenced in the main manuscript.

Response 15. Yes, I think you must include the use of Hill number instead of classical analysis for the reasons that I explained to you last time.

Response16. You did not introduce multivariate analysis but univariate ones which is good. But can you introduce PCA method in the analysis, do you make standardisation of your data ?

I did not see the figure 5...

I am interested to see the listing of the taxa as I asked in the first report but this is still not included in the paper, can you show the list of the specimen that you capture from Guandi Mountains ? This is central to your study because you assessed the gradient of elevation based on the species richness.

Author Response

Thanks so much for reviewer 3 for further modification suggestions to improve the quality of our work. We have revised the manuscript as reviewer mentioned. Those comments are all valuable and very helpful for revising and improving our manuscript, as well as the important guiding significance to our researches. We have studied the comments carefully and have revised and corrected the manuscript which we hope meet his/her approval. Revised portions are marked in light blue in the revised manuscript. The point to point responses are as following:

Point 1 (Response 7): Please note the original reference for the climate zone and not from forest paper. This mean that you can cite reference(s) from public climate data (maybe from china). 

Response 1: Thanks very much for reviewer’s clear advice. The new Climate-related references have been added to L105 for MS, which is the National Meteorological Information Center (NMIC) of China Meteorological Administration (CMA). The new reference is as follows:

  1. The National Meteorological Information Center (NMIC) of China Meteorological Administration (CMA). Available online: http://www. data.cma.cn/.

Point 2 (Response 10): The schema of the sampling design is missing... in the manuscript. 

Response 2: Thanks very much for reviewer’s clear advice. The schema (Figure 1) of “pitfall trapping” sampling design has been added to L129 of the MS, as detailed below:

Figure 1 The schema of “pitfall trapping” sampling design

Point 3 (Response 11): Yes, in the supplemental information, the figure could be added and referenced in the main manuscript.

Response 3: Thanks very much for reviewer’s clear advice. The Figure S1 “Field operation of "pitfall trapping"” has been added in the supplementary materials of the MS. The reference to Figure S1 has also been added to L133 of the MS.

Point 4 (Response 15): Yes, I think you must include the use of Hill number instead of classical analysis for the reasons that I explained to you last time.

Response 4: Thanks very much for reviewer’s clear advice. I would like to thank the reviewers for proposing to use Hill number to solve some problems caused by the analysis of alpha diversity index in order to optimize the quality of MS. According to the comments of the reviewers, we have calculated the Hill number of the insect community changing with the elevation gradient. The Tables 3 “The Hill numbers of insect species composition and diversity in typical vegetation community ecosystems along altitudinal gradient of the Guandi Mountain.” and contents related to Hill numbers (L163-166, L222-226, and L238-241) have been added to MS. In addition, Figure 5 “Difference of insect diversity along elevation gradient in the Guandi Mountain” has been removed from the MS.

Point 5 (Response 16): You did not introduce multivariate analysis but univariate ones which is good. But can you introduce PCA method in the analysis, do you make standardisation of your data ?

Response 5: We are sorry for our negligence in forgetting to include PCA method in MM, and we are very grateful to the reviewer for their useful suggestions. According to reviewer’s advice, the sentence “Principal component analysis (PCA) was used to examine the differentiation characteristics of insect communities along the altitude gradient using a free online platform for data analysis (https:// www. genes cloud. cn)” has been added in L160-163 in the M&M. In addition, it is important and necessary to standardize the data in PCA analysis.

Point 6: I did not see the figure 5...

Response 6: Thanks very much for reviewer’s clear advice. According to the comments of the reviewers, Figure 5 “Difference of insect diversity along elevation gradient in the Guandi Mountain” has been removed from the MS.

Point 7: I am interested to see the listing of the taxa as I asked in the first report but this is still not included in the paper, can you show the list of the specimen that you capture from Guandi Mountains ? This is central to your study because you assessed the gradient of elevation based on the species richness.

Response 7: Thanks very much for reviewer’s clear advice. The Table S3 “List of the collected insects in the Guandi Mountain” has been added in the supplementary materials of the MS.

Response 8: Moreover, in order to further improve the quality of the MS, we have gone through the manuscript again to check the possible problems and made corresponding modifications, as detailed below:

  1. Table S3 “List of the collected insects in the Guandi Mountain” has been added in the supplementary materials of the MS.
  2. “Table S1”, “Table S2” and “Table S3” have been added in L111, L179, and L204 respectively.
  3. Due to the addition and reduction of some references, the order of references in MS has been adjusted to some extent, and we have also marked light blue in MS accordingly.
  4. We have removed the original Figure 5 from MS, therefore, the order of Figures in MS has been adjusted to some extent, and we have also marked light blue in MS accordingly.

The word “number” has been changed to “abundance” in L336 of the M

Round 3

Reviewer 3 Report

Thank to the author for the consideration of the revision that were asked.

For me, it is OK, the only thing is to write each species with the first descriptor after the scientific name in table S3

Author Response

Point 1: Thank to the author for the consideration of the revision that were asked. For me, it is OK, the only thing is to write each species with the first descriptor after the scientific name in table S3

Response 1: We would like to thank reviewer 3 for their valuable revision comments on the quality of MS, which we believe can greatly improve the quality of the paper. According to reviewer’s advice, the first descriptor of insect species (very few of them can't be found) in Table S3 have been added. Please check the supplementary materials. Thank you again for your help in improving the quality of our articles.